# Active Immunization Against Inhibin Impaired Spermatogenesis, Plasma Luteinizing Hormone, Pituitary Prolactin mRNA, and Hypothalamic Vasoactive Intestinal Peptide mRNA Expressions in Yangzhou Ganders

**DOI:** 10.3390/vetsci12050413

**Published:** 2025-04-27

**Authors:** Muhammad Faheem Akhtar, Muhammad Umar, Ejaz Ahmad, Mingxia Zhu, Ying Han, Changfa Wang

**Affiliations:** 1Liaocheng Research Institute of Donkey High-Efficiency Breeding and Ecological Feeding, Liaocheng University, Liaocheng 252059, China; 2Department of Animal Reproduction, Faculty of Veterinary and Animal Sciences, Lasbela University of Agriculture, Water and Marine Science, Uthal 90150, Pakistan; umar.vas@luawms.edu.pk; 3Department of Clinical Sciences, Faculty of Veterinary Science, Bahauddin Zakariya University, Multan 60800, Pakistan; ejaz.ahmad@bzu.edu.pk

**Keywords:** INH immunization, testicular histoarchitecture, PRL mRNA, VIP mRNA, Yangzhou ganders

## Abstract

Reproductive efficiency in the goose is an innovative approach critical for the thriving global industry. A poor reproductive performance and seasonality in reproduction are two of the most significant challenges in optimizing reproductive efficiency in the goose industry. Therefore, maximum profits from the goose breeder stock cannot be gained without achieving higher fertility and hatchability. Inhibin (INH) is a key hormone secreted by Sertoli cells in the testes, mainly involved in the regulation of follicle-stimulating hormone (FSH) and luteinizing hormone (LH) production. Furthermore, INH immunization has elevated reproduction outcomes in some mammalian and commercial avian species. Our results demonstrate that INH immunization significantly affected spermatogenesis and seminiferous epithelium. LH concentrations (ng/mL) were observed to be lower in the INH-immunized group compared with the control group. Hypothalamic vasoactive intestinal peptide (VIP) mRNA and pituitary prolactin (PRL) mRNA expressions remained higher in the control group at 181 and 200 days of age in both, except on day 227. Overall, the data suggest that the INH immunization did not improve reproductive efficiency in Yangzhou ganders concerning plasma LH, testicular histology, pituitary PRL mRNA, and hypothalamic VIP mRNA expressions.

## 1. Introduction

Males, especially male geese, have reproductive physiology that is regulated by the glycoprotein hormone inhibin. It is mostly released by the Sertoli cells and is important for the feedback control of the pituitary gland’s release of follicle-stimulating hormone (FSH). There are two primary forms of inhibin, inhibin A and inhibin B. Inhibin B is more common in men and is a marker for spermatogenic status and Sertoli cell function [1]. Spermatogenic activity in the testes is intimately related to inhibin B secretion. According to studies, inhibin B levels have an inverse relationship with FSH levels and a positive correlation with sperm count and Sertoli cell activity [2]. This feedback system is essential for preserving the gonadotropin balance required for healthy spermatogenesis. Age and metabolic state are two variables that affect inhibin B production; decreases in inhibin B levels are frequently indicative of decreased spermatogenesis [3]. The expression of inhibin subunits has been observed in the testes at different developmental stages, suggesting a role in the maturation process of the male reproductive system [4]. Inhibin’s role extends beyond simple feedback inhibition, as it also is involved in the paracrine regulation of testicular function, influencing the activity of Leydig cells and the overall hormonal milieu within the testes [5].

Studies on other bird species, including Japanese quail, have shown that inhibin is released from the testes and plays a role in controlling gonadotropin levels, which suggests that inhibin is physiologically significant in male geese [6]. Active immunization against INH improved testes weight and plasma testosterone concentrations in aging broiler breeders [7]. Another very important modulator of fertility is a VIP [8]. VIP functions as a neurotransmitter. It is not only produced in the hypothalamus but also distributed in the testis and involved in gametogenesis and testosterone synthesis [9,10,11,12,13,14]. In seasonal breeders, seasonality has a marginal effect on ganders’ reproductive efficiency. Seasonal reproduction in both short and long-breeding birds is regulated by gonadotrophin-releasing hormone (GnRH) and VIP in the hypothalamus, equivalent to PRL and LH in the pituitary gland [15,16].

In goose production, the reproductive efficiency of ganders declines with seasonal breeding [17]. After two months of good reproductive efficiency, a decline in semen quality and fertility, accompanied by alterations in endocrine parameters and sexual behaviors, was observed in ganders [18,19]. In commercial goose production, this evidence raises attention to the importance of a high reproductive male breeder stock. It is the need of the hour to find the reasons for the poor reproduction of ganders and remedies explored by poultry endocrinologists and theriogenologists all over the world to improve commercial gander stock reproduction. In recent years, there have been attempts to enhance reproductive efficiency in ganders by changing lighting regimes, breeding stage, dietary supplementation of vitamin E and selenium, and crossbreeding [20,21]. INH immunization is one of the optimal tools to improve reproductive efficiency in birds and mammals. In birds, both LH and PRL are secreted by the anterior pituitary gland. LH acts on Leydig cells to produce testosterone (T), while FSH acts on Sertoli cells that secrete inhibin. Inhibin via the feedback loop acts on FSH, and Leydig cells also need inhibin for testosterone (T) synthesis, as inhibin acts as a connection between Leydig and Sertoli cells for spermatogenesis. This makes a cross-link among LH, PRL, and inhibin. FSH is an important modulator in this loop. So, we hypothesized that active immunization against INH may improve pituitary PRL and hypothalamic VIP and, thus, it may aid in improving the reproduction of Yangzhou ganders. All geese that breed in China, except Yili geese, developed from wild swan geese (*Anser cygonoides*) approximately 6000 years ago [22]. Our experimental bird, the Yangzhou goose, is a long-day breeder bird of Chinese origin developed using local germplasm resources. It is popular due to its good fertility, hatchability, egg, and carcass quality. LH in the pituitary gland and VIP are important players in the hypothalamus–pituitary–gonadal axis (HPG axis) and reproductive cycle [23,24]. Vasoactive intestinal peptide is secreted in the hypothalamus and inhibits the reproductive performance of young White Leghorn roosters [25]. Similarly, INH inhibits the secretion of FSH; so, the activity of these two hormones, i.e., VIP and INH, is interlinked with the hypothalamus–gonadal axis (HPG). In our previous work [26], we discovered that INH immunization affects spermatogenesis and testicular development in Yangzhou ganders. Also, Sertoli cell genes (SOX9, Wt1, Dhh, AMH, and FSHR) were downregulated with INH immunization, implying that Sertoli cell development was downregulated. Also, we described that INH immunization regresses germ cell (spermatogonia and spermatocytes) development in the seminiferous epithelium of Yangzhou ganders. Our previous findings show that INH immunization has an effect on the hypothalamus–pituitary–gonadal axis, but whether it affects LH, PRL, and PRL mRNA expressions in the pituitary gland and hypothalamus remains unknown. Molecular regulation of inhibin and related hormones on hypothalamus pituitary gonadal axis is shown in Figure 1. The purpose of the present study was to elucidate the impact of INH immunization on LH changes in the seminiferous epithelium and the relative mRNA expressions of PRL and VIP in the pituitary gland and hypothalamus. 

## 2. Materials and Methods

### 2.1. Ethical Considerations

This experiment was conducted at the Nanjing Agricultural University, China. All experimental designs and procedures were approved regarding ethical conditions by the Animal Care and Use Committee (ACUC) held at Jiangsu Academy of Agriculture Sciences, Nanjing Agricultural University, China. Vide reference nos. 31572403 and 31402075.

The present study was conducted using prepared INH protein injected in Yangzhou ganders used as an experimental model. The whole study was completely conducted as a blind (masking) evaluation. The method of protecting the study group by blind evaluation after randomization is usually used to reduce the risk of bias in clinical trials.

### 2.2. Place of Experiment

The current experiment was conducted at Sunlake Swan farm in Henglin Township, Changzhou, Jiangsu province, China.

### 2.3. Selection of Birds and Experimental Design

A total of 60 Yangzhou ganders (having the same genetic origin) were selected for this study and randomly assigned into two groups, A and B. These birds were kept under the naturally prevailing climatic conditions and were allowed to acclimatize for one week before the start of the experiment. For identification, tags were attached under the wings of each bird. Each group comprised 30 birds; group A was kept as the control (CON), while group B was kept as the INH group. Birds in CON were immunized with BSA (bovine serum albumin), while those in the INH group were immunized against INH. Throughout the study, the birds were housed at temperatures ranging from 25 °C to 32 °C. The ganders had free access to water and were fed a diet containing 12.5% crude protein (CP), which was mixed with grass. Ad libitum feed was administered in the day time (from sunrise to evening). The mean body weight (BW) values of the control and INH-immunized groups are provided in Table 1.

### 2.4. INH Preparation

A recombinant goose INH fusion peptide (142 amino acid residue), plasmid pRSET-A, and a 106-residue fragment of goose INH α-subunit mature peptide fragment were expressed in *Escherichia coli* BL21(DE3) and purified according to the protocol described by [27]. After the final purified concentration of INH protein was 3 mg/mL, INH protein was then mixed with a mineral oil adjuvant (Solarbio Life Sciences, San Diego, California, USA) at a 1:2 (*v/v*) ratio. The INH protein and mineral oil adjuvant mixture was condensed to achieve a final protein concentration of 1 mg/mL, preparing the INH immunogenic protein. Subsequently, physiological saline was mixed with the mineral oil adjuvant to prepare the bovine serum albumin (BSA) solution. Ganders in the INH group were immunized with the INH protein on days 161, 181, and 209, while birds in the CON group were inoculated with BSA on the same days. The first immunization occurred when the birds were 161 days of age. It typically takes 20 days for the inhibin vaccine to induce an antibody response and produce antibodies. Therefore, the second booster shot was administered at 181 days of age, and the third inhibin shot was administered at 209 days of age. Consequently, time points at 181, 200, and 227 days of age were selected for sampling. Additionally, by 161 days of age, Yangzhou ganders begin to reach their optimal body weight (B.W.), which further justified the selection of 181, 200, and 227 days to assess whether age has any impact on the immune response.

### 2.5. Tissue Sample Collection

Hypothalamus and pituitary tissues were collected at 181, 200, and 227 days of age. At 181 days of age, 10 birds from each group were selected and euthanatized for tissue sampling. Similarly, 10 birds from each group were randomly selected and euthanatized on days 200 and 227 of age for tissue collection. The samples were immediately stored at −80 °C after collection. Blood samples were obtained through brachial venipuncture into heparinized tubes on days 161, 181, 190, 200, 225, and 227 (5 mL, Hangzhou Rollmed Co., Ltd., Hangzhou, China). The heparinized tubes were centrifuged at 2000× *g* for 3 min at 4 °C. Within 3 h of blood collection, the plasma was separated by centrifugation at 1000× *g* and stored at −20 °C.

### 2.6. LH Plasma Concentration Measurement

Plasma LH concentration was quantified using chicken LH (USDA-cLH-K-3) as the reference standard and USDA-cLH-I-3 as the radiolabeled ligand. Rabbit anti-chicken LH (USDA-AcLH-5) and donkey anti-rabbit antiserum (Abcam ab7080, Cambridge, MA, USA) were used as the primary and secondary antibodies, respectively. The intra-assay and inter-assay coefficients of variation were both below 15%, and the sensitivity of the assay was 0.1 ng/mL. It should be noted that the R-values of the standard curves were consistently above 0.99.

### 2.7. Complementary DNA (cDNA) Synthesis and Quantitative Real-Time PCR (qRT-PCR)

RNA was extracted from the pituitary gland and hypothalamus using the Trizol reagent with an RNA extraction kit (Tianjin, China), following the manufacturer’s instructions. Briefly, 0.2 mL of chloroform (Henan GP Chemicals, Zhengzhou, China) was added to the Trizol reagent, and the mixture was homogenized by shaking. The sample was then centrifuged at 12,000× *g* for 15 min at 4 °C. The aqueous phase (supernatant) was mixed with 0.5 mL isopropanol and centrifuged again at 12,000× *g* for 10 min at 4 °C. The resulting RNA pellet was washed with 75% ethanol, air-dried, and re-suspended in 20 µL of sterile treated diethylpyrocarbonate water.

RNA concentration and purity were assessed using a spectrophotometer (NanoDrop 2000c, Thermo Scientific, Dreieich, Germany). For reverse transcription, 5 µg of RNA was used with the Takara PrimeScript™ RT Reagent Kit (RR037, Takara, Osaka, Japan) to synthesize complementary DNA (cDNA). Gene sequences are listed in Table 2. The entire process of RNA isolation, cDNA synthesis, and quantitative real-time PCR (qRT-PCR) followed the protocol described in [28]. Gene expression levels were calculated using the 2^−ΔΔCT^ method.

### 2.8. Microscopy

To observe histological alterations in the seminiferous tubules, a small portion (0.125 cm^3^) of the left testis tissue was collected and fixed in 10% buffered formalin for 24 h. Histological evaluation was performed using a tissue processor (LEICA RM 2235). Testicular tissues underwent dehydration in ascending concentrations of alcohol (70%, 80%, 90%, and 100%), followed by absolute alcohol. After clearing in xylene, the tissues were embedded in melted paraffin wax.

Sections of 5 µm thickness were mounted onto glass slides. The slides were then stained with hematoxylin and eosin. These tissue sections were analyzed under a light microscope (Olympus BX63) at magnifications of 10× and 40× to examine changes in spermatogonia, spermatocytes, seminiferous tubule (ST) diameter, and elongated spermatids.

### 2.9. Statistical Analysis

Statistical analysis was performed using SPSS (Version 20.0, Armonk, NY, USA) and GraphPad Prism (Version 5.0). The Kolmogorov–Smirnov test was applied to assess the normality of the data. If the data were not normally distributed, they were log-transformed and re-tested for normality before further analysis. A two-way ANOVA was performed to compare mean values, with the results expressed as mean ± SEM, and group differences at experimental time points were analyzed using the Bonferroni post-test, considering *p* < 0.05 as statistically significant.

## 3. Results

### 3.1. Plasma LH Concentrations

Plasma hormone concentration in both groups (CON and INH) showed similar ascending and descending patterns with few exceptions throughout the experimental duration. After the first INH immunization on day 161, plasma LH concentrations declined from 2.5 (ng/mL) to (1 ng/mL) in the CON group and from 2 (ng/mL) to 1 (mg/mL) in the INH group. After the second INH immunization on day 181, plasma LH concentrations increased to 2 (ng/mL) in both groups on day 190. Then, it showed a declining pattern on day 200. The plasma LH concentration significantly increased on day 225 in both groups A and B (CON and INH) after the third booster of INH on day 209. Similarly, the plasma concentration of the CON group showed a slight elevation until day 227 of the post-immunization treatment; however, the plasma LH concentration had the lowest levels in group B (INH-treated). Moreover, the plasma concentration of LH in both groups A and B (CON and INH) followed similar patterns of ascending and descending until day 225 of age. Plasma LH concentrations in both groups CON and INH are shown in Figure 2a.

### 3.2. Pituitary PRL and Hypothalamic (VIP) mRNA Expression

The quantitative real-time (qRT-PCR) analysis showed that the PRL mRNA expression was reduced on day 181 compared to the CON group of Yangzhou ganders. The statistical analysis of PRL mRNA expression showed a significantly decreased pattern on day 200 after the second dose of immunization in Yangzhou ganders of the INH group. In contrast, at day 227 of age, mRNA expression was increased in the INH group after the third dose of immunization (Figure 2b).

The quantitative real-time (qRT-PCR) analysis showed that the level of VIP mRNA expression in the hypothalamus was significantly (*p* < 0.001) decreased in the INH group, while significant (*p* < 0.001) increases were observed in the INH group at day 181 of age after the immunization treatment. However, the level of VIP mRNA expression in the hypothalamus was increased in the INH group, and a lower expression was observed in the CON group after the immunization treatment at the end of the experiment at day 227 of age (Figure 2b).

### 3.3. Histology of Testes

Towards the end of the study, the histological findings of the testes of the INH group were significantly different from the (CON) group after the treatment of immunization at days 181, 200, and 227 of age. Additionally, normal development of germ cells was observed in both groups post-treatment of INH immunization at days 181, 200, and 227. However, the number of germ cells in histological sections in the INH group appeared deficient as compared to the CON group.

Histologically, testes collected from Yangzhou ganders in the CON group were significantly filled with spermatogonia, spermatocytes, and spermatids. However, the histological study of the INH-immunized group showed irregular shrinkage of seminiferous tubules with a marked reduction in the number of germ cells, including spermatogonia, spermatocytes, and spermatids. In Figure 3, the histological photomicrograph shows morphological alterations in the seminiferous tubules of the testes of Yangzhou ganders after the post-INH immunization treatment on days 181, 200, and 227.

## 4. Discussion

The effects of INH immunization on Yangzhou ganders’ plasma LH levels, histological testicular alterations, and PRL and VIP mRNA expressions in the pituitary gland and hypothalamus are complex and can be explained by hormonal interactions and regulatory systems.

In the hypothalamus–pituitary–gonadal (HPG) axis, INH plays a critical role [28]. To improve spermatogenesis and reproductive efficiency, mammalian species have been immunized against INH [29]. In Merino rams, sperm counts began to rise following INH vaccination [30]. Our results show a significant difference in LH concentrations between the two groups on day 225, following the third dose of INH immunization. Moreover, the findings of the present study align with those of [31], which found no significant effect of inoculation with the INH-α subunit on plasma LH concentrations in developing cockerels. Similarly, in Shiba goats, passive INH immunization did not affect LH concentrations [32]. FSH acts on Sertoli cells and seminiferous tubules in initiating spermatogonium division [33], while LH influences Leydig cells to produce testosterone. This suggests that both LH and FSH play crucial roles in spermatogenesis [34].

Gonadotropin-releasing hormone (GnRH) and INH, directly or indirectly, influence LH and FSH regulation in the hypothalamus–pituitary–gonadal axis [35]. In agreement with our findings, it was reported that a significant declining pattern of LH concentrations was observed in the INH group compared to the rest of the experimental CON group. Moreover, by neutralizing endogenous INH, INH immunization has been suggested to increase FSH concentrations and stimulate follicular development [36]. It shows that only INH immunization does not seem to modulate endocrinological changes, and simultaneously, the photoperiod has its effect on the modulation of LH. The geographical location affects the annual pattern of seasonality and reproduction in domestic geese [37]. In various avian species, the photoperiod is a central factor for seasonal reproduction control in the temperate zone [38,39]. Yangzhou geese are long-day breeding geese species whose breeding period starts in October, peaks in February–March, and ends in June. The age of maturity largely affects the reproductive activities of birds, not only the seasonality [40]. Until Yangzhou ganders reach the age of maturity, the major factor, i.e., photoperiod, peaks, modulating LH plasma concentrations followed by similar T concentrations. In contrast, high levels of T have a positive feedback effect on male animal spermatogenesis and maturation, but they also negatively affect LH levels [41].

In our results, testicular histology showed a reduced rate of spermatogenesis in the INH-immunized group compared to the control group after INH immunization. Our findings are as those presented in [42], in which INH lowered spermatogonial numbers in the testes of mature mice and Chinese hamsters. According to a prior study, Yangzhou ganders’ testicular cell counts began to decline following INH immunization; testicular apoptosis may be caused by the combined effects of INH and seasonality [43]. Moreover, seasonality in breeding improves the rate of apoptosis, which is a typical event in proliferating and continuously regenerating tissues and seminiferous epithelium [44]. In cockerels, the size of the testicles has a direct impact on sperm production [45].

Sertoli cell expansion is indicated by am improved mRNA expression of Sertoli cell genes [46]. In the current investigation, the expression patterns of pituitary PRL mRNA and hypothalamus VIP mRNA were nearly identical, with very few exceptions. A prior study found similar expressions of PRL mRNA and VIP mRNA [47]. Moreover, the INH immunization increased the *FSHR* mRNA in the large white follicles of partridge shank hens. LH-β mRNA expression remained unaffected after INH immunization [48]. Pituitary PRL mRNA expressions were markedly elevated in this study and completely at odds with the LH plasma concentration. Similar results of an inverse connection between plasma PRL and LH and T concentrations in Zatorska ganders during the first reproductive cycle were previously reported by other researchers. We can speculate from these results that PRL may influence endocrinological variations in LH plasma concentrations, which are then reflected in pituitary PRL mRNA expressions, leading to a decreased rate of spermatogenesis. Moreover, PRL appears to influence changes in the first reproductive cycle, such as LH plasma concentration, which may change during the second season of reproductive activity. It is necessary to clarify the direct effect of PRL on testicular morphology.

## 5. Conclusions

Yangzhou ganders’ reproductive activities are regulated in the HPG axis by the breeding season and their maturity age. An important part of the hypothalamus–pituitary–gonadal (HPG) axis is played by the daily photoperiod and INH immunization. INH immunization caused disruptions to testicular histology and germ cells, resulting in decreased spermatogenesis efficiency in Yangzhou ganders. However, INH immunization may harm pituitary PRL mRNA, hypothalamic mRNA expressions, and LH plasma concentration. Moreover, Yangzhou geese are long-day breeding birds; even following INH immunization, the photoperiod has a significant impact on reproductive and endocrinological changes in Yangzhou geese. To gain a deeper understanding, more molecular pathways must be investigated.

## Figures and Tables

**Figure 1 vetsci-12-00413-f001:**
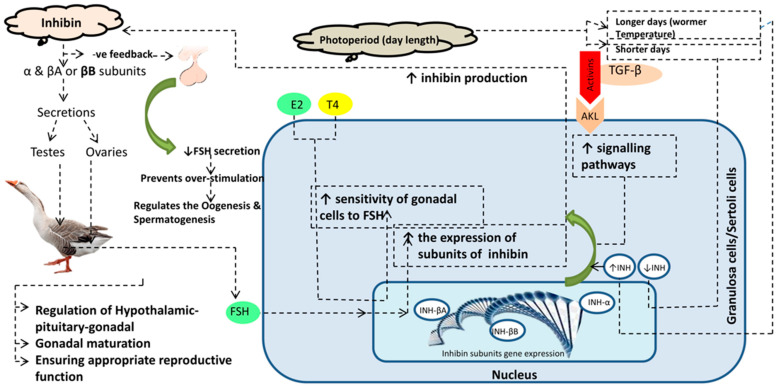
Graphical abstract showing the molecular regulation of inhibin and related hormones on the hypothalamus–pituitary–gonadal axis. As seasonal breeder birds (long and short days), the photoperiod has a direct effect on hormonal regulation and the HPG axis. Inhibin protein is produced by Sertoli cells (having subunits α and β). Elevated inhibin (INH) produces a negative feedback in relation to FSH and lowers its secretion.

**Figure 2 vetsci-12-00413-f002:**
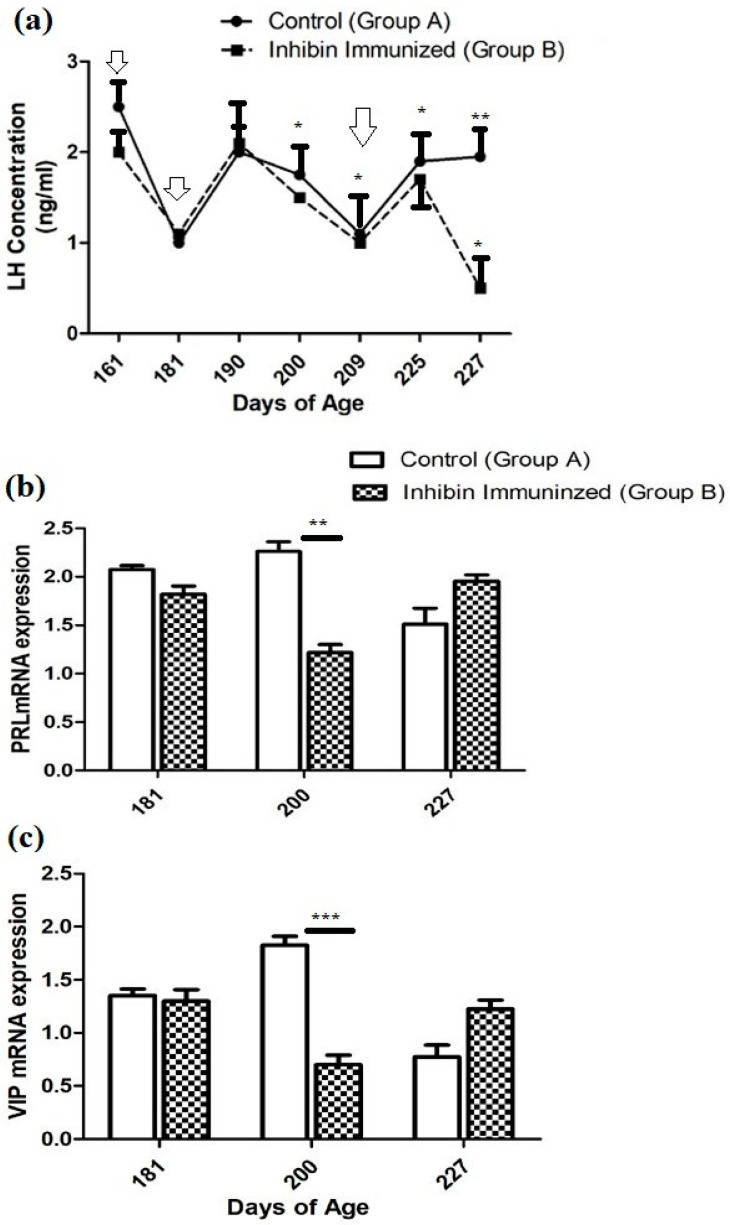
(**a**) Plasma LH concentrations in the INH (INH) and control (CON) groups. Yangzhou goose ganders at ages 161, 181, 190, 200, 225, and 227 days, with vertical bars indicating the standard error of mean (SEM). The values with ** indicate the difference (*p* < 0.001). Arrows indicate the age of the birds at the time of immunization and following booster shots. (**b**) Pituitary PRL and (**c**) hypothalamic VIP mRNA levels in the INH (INH) and (CON) groups of Yangzhou goose ganders. Data are shown as mean values ± standard error of the mean. *, **, and *** indicate differences at *p* < 0.05, *p* < 0.01, and *p* < 0.001, respectively, among the groups.

**Figure 3 vetsci-12-00413-f003:**
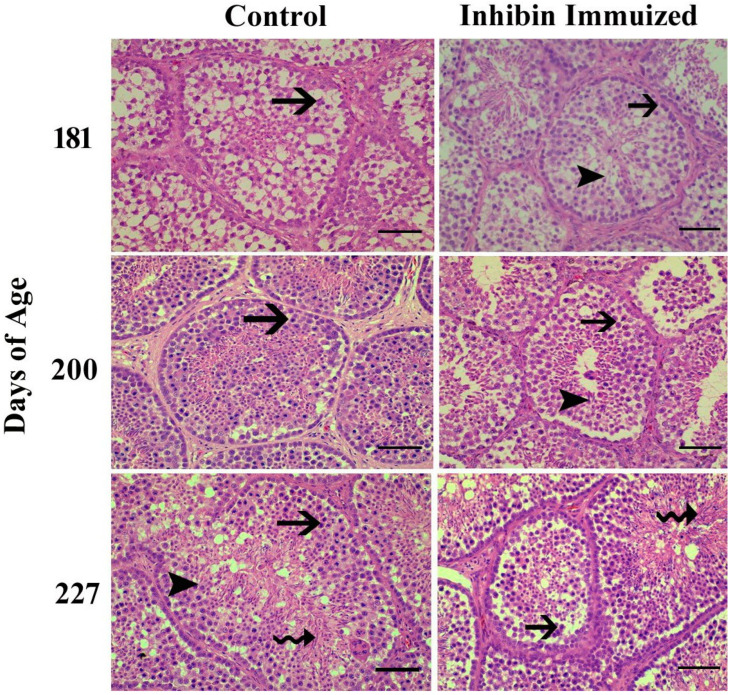
Photomicrographic testicular changes in Yangzhou ganders at 181, 200, and 227 days of age in the control and INH-immunized groups. Arrows with tails show spermatogonia; arrowheads show spermatogonia; and arrows with spiral tails show spermatids. The scale bar represents 20 μm at 40× magnification.

**Table 1 vetsci-12-00413-t001:** Mean body weight (BW) of Yangzhou ganders.

Day of Age	Control (Kg)	INH Immunized (kg)
181	4.73	4.68
200	4.41	4.54
227	4.56	4.88

**Table 2 vetsci-12-00413-t002:** Primers used in real-time quantitative PCR (qRT-PCR) for gene transcription.

Gene Name	Sequences (5′→3′)	Product Length (bp)	Tm (°C)	Accession Number
VIP	upstream: ACCAGTGTCTACAGCCATCTTTG downstream: AGGTGGCTCAGCAGTTCATCTACA	1446	-	106034217
PRL	upstream: TGCTCAGGGTCGGGGTTCA downstream: GCITGGAGTCCCATCGGCAAGTT	218	56	DQ023160
β-actin	upstream: TGACGCAGATCATGTTGAGA downstream: GCAGAGCGTAGCCCCATAG	159	60	AY149895

## Data Availability

The datasets analyzed during the current study are not publicly available due to the individual privacy of reindeer owners, but are available from the authors upon reasonable request.

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
