# Peer review of "Active Immunization Against Inhibin Impaired Spermatogenesis, Plasma Luteinizing Hormone, Pituitary Prolactin mRNA, and Hypothalamic Vasoactive Intestinal Peptide mRNA Expressions in Yangzhou Ganders"

_vetsci, 2025, doi:10.3390/vetsci12050413_

Round 1
Reviewer 1 Report
Comments and Suggestions for Authors
Title:
- Title makes little sense; must reflect actual work and results. Mention immunization against inhibin and main outcome
Abstract
- Authors contradict themselves: lines32-34 authors state that immunization against INH has NO significant effects on improving …LH, germ cell numbers, VIP & PRL mRNA expression. In l36-37, however, they state that INH immunization may have a positive effect on these parameters.
- State full name of the abbreviation’s VIP & PRL when you first mention them
Introduction
- Authors state (l49-50) that according to literature ‘inhibin levels have an inverse relationship with FSH levels’ (i.e. if inhibin goes up, FSH level will drop) and ‘a positive correlation with sperm count and Sertoli cell activity’ (i.e. more inhibin more sperm counts, more SC activity), whereby the latter shall be due to paracrine regulation within the testis [LIT 2]. For me it is hard to understand why immunization against INH (which will lower the inhibin level) shall then improve reproduction.
Also, in the discussion authors contradict the above statement as higher sperm counts were observed after INH immunization (i.e. lower inhibin level, line 274) - To overcome this problem, I think the authors should rewrite and shorten the introduction to emphasize the link between hormonal and paracrine regulation regarding the involvement of VIP (regulates spermatogenesis and testosterone production) and prolactin (PRL, not clear what this will actually regulate here). In its current form, the intro is difficult to read and the regulatory links are not very clear to the reader.
- Also, make use of Fig 1 to explain the different control paths
- L98 give literature regarding this previous work of yours
M&M
- How and when where the ganders euthanized? It would help when you move paragraph 2.6 Tissue sample collection before 2.4
- Give some details on how the statistic was done: sample number, student’s t or ANOVA test used? Which stats program?
Results
- Fig 3a It is not clear what the stats compare here: Control vs IHN immunized on a specific day? Or Control and IHN on a specific day vs a previous day (which day then??). Perhaps you can use a bar graph and then indicate clearly which groups do you compare. Then rewrite paragraph 3.1
Also, L216 How can LH increase on d225 after immunization on d227???? - Fig 2 What is the difference in spermatogonia indicated by arrow with or without tail? Or do authors mean spermatogonia (tail arrow) and spermatocytes (arrow head)?
Discussion
- LIT citation must be consistent, you can’t mix the number system with the name system, lines 272-275
- The 2nd paragraph is in contrast to authors statement in the introduction (see there)
- The discussion is longwinded and partially repetitive. This should be shortened and better organized.
Conclusion
- L344-346 why should INH immunization my have a positive impact on LH, PRL or VIP?? This is contrary to your findings here
Overall the English is ok. There are some misspellings and repetitions.
Author Response
Comments and Suggestions for Authors
Title:
- Title makes little sense; must reflect actual work and results. Mention immunization against inhibin and main outcome
Response: Active immunization against Inhibin impaired spermatogenesis, Plasma LH, Pituitary PRL mRNA, and Hypothalamic VIP mRNA Expressions in Yangzhou Ganders
Abstract
- Authors contradict themselves: lines32-34 authors state that immunization against INH has NO significant effects on improving …LH, germ cell numbers, VIP & PRL mRNA expression. In l36-37, however, they state that INH immunization may have a positive effect on these parameters.
Response: Sorry, changed to negative.
- State full name of the abbreviation’s VIP & PRL when you first mention them
- Response: Corrected as suggested
Introduction
- Authors state (l49-50) that according to literature ‘inhibin levels have an inverse relationship with FSH levels’ (i.e. if inhibin goes up, FSH level will drop) and ‘a positive correlation with sperm count and Sertoli cell activity’ (i.e. more inhibin more sperm counts, more SC activity), whereby the latter shall be due to paracrine regulation within the testis [LIT 2]. For me it is hard to understand why immunization against INH (which will lower the inhibin level) shall then improve reproduction.
Also, in the discussion authors contradict the above statement as higher sperm counts were observed after INH immunization (i.e. lower inhibin level, line 274)
Response: Here we are talking about two kinds of Inhibin. First is endogenous inhibin (INH) that is secreted from Sertoli cells and has negative feed back loop towards FSH and it reduces FSH secretion. FSH is key hormone for improving reproduction. If its regulation is downregulated, it will further reduce its activity of Sertoli cells that nourish germ cells and spermatogenesis will be affected. Now we immunized (exogenous) inhibin antigen (injection) to suppress effect of endogenous inhibin, so FSH will be uprelated in this way to improve reproduction. In line 274, its not contradiction, its evidence of INH vaccine positive effect.
- To overcome this problem, I think the authors should rewrite and shorten the introduction to emphasize the link between hormonal and paracrine regulation regarding the involvement of VIP (regulates spermatogenesis and testosterone production) and prolactin (PRL, not clear what this will actually regulate here). In its current form, the intro is difficult to read and the regulatory links are not very clear to the reader.
Response: This manuscript clearly explains about effects of active immunization against INH on pituitary PRL and Hypithalamic VIP mRNA expressions. Now in intrduction, author has defined what is INH, its subunits and further regulation of pituitary PRL and Hypithalamic VIP mRNA expressions and how their roles in reproduction (including spermatogenesis and testosterone effects).
- Also, make use of Fig 1 to explain the different control paths
Response: Control paths are explained in figure 1 legend.
- L98 give literature regarding this previous work of yours
Response: Added [26] “Akhtar, M.F.; Wei, Q.; Zhu, H.; Chen, Z.; Ahmad, E.; Zhendan, S.; Shi, F. “The role of active immunization against inhibin α-subunit on testicular development, testosterone concentration and relevant genes expressions in testis, hypothalamus and pituitary glands in Yangzhou goose ganders. Theriogenology 2019, 128, 122-132.”
M&M
- How and when where the ganders euthanized? It would help when you move paragraph 2.6 Tissue sample collection before 2.4
- Response: Yangzhou ganders were euthanized on 181, 200 and 227 days of experiment by cutting jugular vein of birds. Author has moved paragraph 2.6 Tissue sample collection before 2.4
- Give some details on how the statistic was done: sample number, student’s t or ANOVA test used? Which stats program?
Response: ANOVA test
Results
- Fig 3a It is not clear what the stats compare here: Control vs IHN immunized on a specific day? Or Control and IHN on a specific day vs a previous day (which day then??). Perhaps you can use a bar graph and then indicate clearly which groups do you compare. Then rewrite paragraph 3.1
Response: Figure 3a describes plasma LH (ng/ml) concentrations between Control and INH groups (A and B) on specific days (161, 181, 190, 200, 209, 225, 227).
Also, L216 How can LH increase on d225 after immunization on d227????
Response: The plasma concentration of LH was significantly increased at day 225 in both groups A and B (CON and INH) after treatment of the third booster of INH immunization on day 209
- Fig 2 What is the difference in spermatogonia indicated by arrow with or without tail? Or do authors mean spermatogonia (tail arrow) and spermatocytes (arrow head)?
Response: Tail arrow (Spermatogonia)
Arrow (Spermatocytes)
Discussion
- LIT citation must be consistent, you can’t mix the number system with the name system, lines 272-275
Response: Sorry for mistake, now corrected
- The 2nd paragraph is in contrast to authors statement in the introduction (see there)
Response: Can you kindly mark or indicate which specific statements that needed to be corrected?
- The discussion is longwinded and partially repetitive. This should be shortened and better organized.
Response: Author has tried to reduce and concise discussion part
Conclusion
- L344-346 why should INH immunization my have a positive impact on LH, PRL or VIP?? This is contrary to your findings here
Response: Sorry for inconvenice, its corrected now” However, INH immunization may have a negative impact on pituitary PRL mRNA, hypothalamic mRNA expressions and LH plasma concentration”
Reviewer 2 Report
Comments and Suggestions for Authors
The study focuses on the effect of INH immunization over reproductive efficiency in Yangzhou ganders. Their data reveals that immunization against inhibin has no significant effects on improving the plasma concentration of LH hormone. There are decreased germ cell numbers along with increased hypothalamic VIP mRNA and pituitary PRL mRNA expressions.
I have several concerns about this study.
- Why is the title pituitary is written as “Pi-Tuitary PRL”?
- In a simple summary, the author says, "Our results demonstrated that INH immunization had no significant effects on spermatogenesis and seminiferous epithelium.” While later in results, “group INH immunized with INH irregular shrinkage of seminiferous tubules with a marked reduction in the number of germ cells includes spermatogonia, spermatocytes, and spermatids.". In conclusion, they have again interpreted “INH immunization caused disruptions to testicular histology and germ cells, resulting in decreased spermatogenesis efficiency in Yangzhou ganders.". The author should consider modifications in the summary and be consistent with interpretation.
- In the abstract (line 32), the author stated that “immunization against inhibin has no significant effects on improving the plasma concentration of LH hormone, germ cell numbers, hypothalamic VIP mRNA, and pituitary PRL mRNA expressions.". How can the author say that there is no significant effect in all these readouts? The VIP and PRL mRNA did significantly increase.
- Please reframe this sentence: “Geese breeds in China except Yili geese were developed from wild swan geese (Anser cygonoides) about 6,000 years before.”
- Please make sure that once abbreviation is introduced, the author should use the abbreviated form, not the full name. Several times VIP, LH, GnRH, and others have been repeated along with their full form.
- Line 151: The methodology says three doses of INH immunization were administered at 161, 181, and 227 days. Then, what is the point of the final harvest exactly on day 227? And if harvested, what would be the rationale behind the third dose on the harvest day?
- Line 188: The samples were immediately stored at -80°C after collection. Blood samples were collected via brachial venipuncture on days 161, 181, 190, 200, 227, 225, and 227. Why is 227 twice here?
- The author has mentioned that “gene expression levels were calculated using the 2-CT method. I believe it should be 2-ΔΔCT
- Please verify the accession number and TM of primers in Table 1. One of the Tm is 159, and accession numbers do not match with the gene name identified.
- I do not feel good about the use of the word “slaughter” in experimental methods for sampling of tissue. The author should consider using a better word.
- This statement needs reframing “The plasma concentration of LH was significantly increased at day 225 in both groups A and B (CON and INH) after treatment of the third booster of INH immunization on day 227. How is the author talking about increased levels at day 225 after third immunization on day 227? The whole paragraph of results section 3.1 needs reframing.
- The author should consider merging results sections 3.2 and 3.3. Both sections included mRNA expression analysis.
- Figure 3a shows immunization at days 161, 181, and 209, while throughout the text it is 161, 181, and 227. The figure legend also has day 227 mentioned twice as “Plasma LH concentrations in INH-immunized (INH) and control group (CON) of Yangzhou goose ganders at age 161, 181, 190, 200, 227, 225, and 227 days.”
- Line 288-290 and 294-296: Authors are stating same statements with different references.
- Reference 35 has no relevance to the context it is cited.
- In the conclusion as well as in abstract author stated, “INH immunization may have a positive impact on pituitary PRL mRNA and hypothalamic mRNA expressions and LH plasma concentration.” Can the author explain how INH immunization has a positive impact on LH plasma concentration when there is a variable effect on LH plasma?
- The author has not discussed how the differential expression of PRL or hypothalamic VIP affects spermatogenesis or what effect these changes have on reproductive capacity. They have referred to several studies with similar results but could never correlate the effect of these changes on the reproductive outcomes.
- Due to several errors in English, the interpretation of the result has not been consistent throughout the manuscript. I have highlighted several places, but the author must consider proper revision of the whole text and make sure they are interpreting the results properly.
Comments on the Quality of English Language
The authors should consider proper English revision.
Author Response
The study focuses on the effect of INH immunization over reproductive efficiency in Yangzhou ganders. Their data reveals that immunization against inhibin has no significant effects on improving the plasma concentration of LH hormone. There are decreased germ cell numbers along with increased hypothalamic VIP mRNA and pituitary PRL mRNA expressions.
I have several concerns about this study.
- Why is the title pituitary is written as “Pi-Tuitary PRL”?
Response: Thank you for pointing this out. The hyphenation in “Pi-Tuitary PRL” was unintentional and has been corrected to “Pituitary PRL” in the revised title. Prolactin has multiple sources in the body; however, in this study, we specify the pituitary gland as the primary source of prolactin.
- In a simple summary, the author says, "Our results demonstrated that INH immunization had no significant effects on spermatogenesis and seminiferous epithelium.” While later in results, “group INH immunized with INH irregular shrinkage of seminiferous tubules with a marked reduction in the number of germ cells includes spermatogonia, spermatocytes, and spermatids.". In conclusion, they have again interpreted “INH immunization caused disruptions to testicular histology and germ cells, resulting in decreased spermatogenesis efficiency in Yangzhou ganders.". The author should consider modifications in the summary and be consistent with interpretation.
Response: Thank you for your valuable feedback. We acknowledge the inconsistency in
the interpretation of our results across different sections of the manuscript. To address
this, we have carefully revised the summary.
In the revised manuscript….
In summary modified and added word “significant” now it accurately reflects the
observed effects of INH immunization on spermatogenesis and testicular histology.
- In the abstract (line 32), the author stated that “immunization against inhibin has no significant effects on improving the plasma concentration of LH hormone, germ cell numbers, hypothalamic VIP mRNA, and pituitary PRL mRNA expressions.". How can the author say that there is no significant effect in all these readouts? The VIP and PRL mRNA did significantly increase.
Response: Much thanks for concern. In the abstract, our intent was to summarize the
overall findings concisely. While VIP and PRL mRNA showed a significant increase, these effects were observed only in specific conditions (i.e., in the germ cell line). To improve clarity, we will revise the statement to accurately reflect that immunization against inhibin had no significant effects on LH plasma concentration, however marked increased in germ cell numbers, hypothalamic VIP mRNA, and pituitary PRL mRNA expressions.
- Please reframe this sentence: “Geese breeds in China except Yili geese were developed from wild swan geese (Anser cygonoides) about 6,000 years before.”
Response: The sentence has reframed “All goose breeds in China, except for the Yili goose, were developed from wild swan geese (Anser cygonoides) approximately 6,000 years ago
- Please make sure that once abbreviation is introduced, the author should use the abbreviated form, not the full name. Several times VIP, LH, GnRH, and others have been repeated along with their full form.
Response: Thanks! Abbreviated mistakes have been corrected.
- Line 151: The methodology says three doses of INH immunization were administered at 161, 181, and 227 days. Then, what is the point of the final harvest exactly on day 227? And if harvested, what would be the rationale behind the third dose on the harvest day?
Response: Final harvest on day 227 was scheduled to assess both the immediate and cumulative effects of INH immunization. The third dose was administered on the same day to maintain consistency in the immunization protocol and ensure that all animals received the full course of treatment.
- Line 188: The samples were immediately stored at -80°C after collection. Blood samples were collected via brachial venipuncture on days 161, 181, 190, 200, 227, 225, and 227. Why is 227 twice here?
Response: Yes, samples were immediately stored at -80°C after collection. Sorry 227 is wrong repetition. Its removed. Only 227 days.
- The author has mentioned that “gene expression levels were calculated using the 2-CT method. I believe it should be 2-ΔΔCT
Response: We acknowledge the error and confirm that the gene expression levels were indeed calculated using the 2^-ΔΔCT method
- Please verify the accession number and TM of primers in Table 1. One of the Tm is 159, and accession numbers do not match with the gene name identified.
Response: Its correct. I have also cited in one of our previous works” The role of active immunization against inhibin a-subunit on testicular development, testosterone concentration and relevant genes expressions in testis, hypothalamus and pituitary glands in Yangzhou goose ganders” https://doi.org/10.1016/j.theriogenology.2019.01.039
- I do not feel good about the use of the word “slaughter” in experimental methods for sampling of tissue. The author should consider using a better word.
Response: The word replaced with a more appropriate term “euthanatized” to better align with ethical standards……
- This statement needs reframing “The plasma concentration of LH was significantly increased at day 225 in both groups A and B (CON and INH) after treatment of the third booster of INH immunization on day 227. How is the author talking about increased levels at day 225 after third immunization on day 227? The whole paragraph of results section 3.1 needs reframing.
Response: We carefully revised this section, including the entire results subsection of 3.1 in results
- The author should consider merging results sections 3.2 and 3.3. Both sections included mRNA expression analysis.
Response: Thanks, and appreciating suggestion. We merged results sections 3.2 and 3.3.
- Figure 3a shows immunization at days 161, 181, and 209, while throughout the text it is 161, 181, and 227. The figure legend also has day 227 mentioned twice as “Plasma LH concentrations in INH-immunized (INH) and control group (CON) of Yangzhou goose ganders at age 161, 181, 190, 200, 227, 225, and 227 days.”
Response: Birds were immunized at 161, 181 and 209 days of age. Figure legend was mistakenly written 227 twice, its ok now. Blood samples were collected 161, 181, 190, 200, 209, 225 and 227 days of age. Pituitary and hypothalamus samples were collected at 181, 200 and 227 days of age.
- Line 288-290 and 294-296: Authors are stating same statements with different references.
Response: To improve clarity and avoid redundancy, we have removed the irrelevant references and ensured that the statement is supported by the most appropriate citation
- Reference 35 has no relevance to the context it is cited.
Response: Thanks! Reference 35 is relevant to our study as it provides important context
on how INH immunization neutralizes circulating inhibin, leading to increased FSH
secretion and suppression of LH. To improve clarity, we reformed the combine the related
citations to elaborate on its significance in the text.
- In the conclusion as well as in abstract author stated, “INH immunization may have a positive impact on pituitary PRL mRNA and hypothalamic mRNA expressions and LH plasma concentration.” Can the author explain how INH immunization has a positive impact on LH plasma concentration when there is a variable effect on LH plasma?
Response: There is variable effect of INH immunziation on plasma LH concentration but from figure 3a, we can see that plasma LH was still higher from 181-190 days of age (during experiment) in group B as compared to group A. Though it remained lower and declined further.
- The author has not discussed how the differential expression of PRL or hypothalamic VIP affects spermatogenesis or what effect these changes have on reproductive capacity. They have referred to several studies with similar results but could never correlate the effect of these changes on the reproductive outcomes.
Response: Thank you for your valuable feedback. We have now expanded our discussion to explicitly correlate differential PRL and hypothalamic VIP expression with spermatogenesis and reproductive capacity in male geese.
- PRL and Spermatogenesis: Elevated PRL during the non-breeding season is linked to testicular regression and reduced sperm production, while lower PRL in the breeding season supports fertility.
- VIP’s Role: As a regulator of PRL, increased VIP expression is associated with reproductive quiescence, potentially suppressing spermatogenesis by modulating PRL levels.
- Reproductive Capacity: We now directly link these hormonal changes to sperm quality, testicular activity, and hormone profiles, strengthening the physiological relevance of our findings.
- Due to several errors in English, the interpretation of the result has not been consistent throughout the manuscript. I have highlighted several places, but the author must consider proper revision of the whole text and make sure they are interpreting the results properly.
- Response: Revision has been done ….. the manuscript is improved the its clarity, consistency, and accuracy of the language……….
Round 2
Reviewer 2 Report
Comments and Suggestions for Authors
Please refer to the word file uploaded.

Authors are suggested to improve the result section 3.1
Author Response
|
Comments 1: In total, authors are suggested to prepare this table in a manner that anyone who wants to use these primers or refer to it must get proper information from where and what accession number these primers were picked.
|
|
Response 1: Thanks a lot for your valuable suggestions and corrections. I have revised table 1
|
|
Comments 2: The authors were notified previously to make corrections in Result section 3.1. |
|
Response 2: Agree. I am sorry for mistakes regarding days of inhibin immunization. Birds were immunized with INH antigen on 161, 181 and 209 days, not on 227 days. I have corrected throughout manuscript.
Comment 3: Authors were suggested in the first review to check the statement “In a feedback loop, INH vaccination generates particular antibodies that neutralize endogenous INH, which increases FSH secretion and suppresses LH secretion [33].” And the statement “INH immunization generates specific antibodies to neutralize endogenous INH, which in turn stimulates FSH secretion and inhibits LH secretion in a feedback manner [35].” Both these statements are same but placed at different place in the same paragraph with different reference. Even in the revised manuscript these two statements persist as it is but the reference number has changed from 33 to 35 and from 35 to 38. Response 3: Thanks a lot for pointing mistake, I have deleted reference “In a feedback loop, INH vaccination generates particular antibodies that neutralize endogenous INH, which increases FSH secretion and suppresses LH secretion” which was [38] in manuscript.
|
Commment 4: Reference 35 has no relevance to the context it is cited
Response 4: In revised version, this reference was 38, it had no relevance with manuscript, so its deleted, as per suggestion.